# Quaternized Amphiphilic Block Copolymers as Antimicrobial Agents

**DOI:** 10.3390/polym14020250

**Published:** 2022-01-08

**Authors:** Chih-Hao Chang, Chih-Hung Chang, Ya-Wen Yang, Hsuan-Yu Chen, Shu-Jyuan Yang, Wei-Cheng Yao, Chi-Yang Chao

**Affiliations:** 1Department of Orthopedics, National Taiwan University Hospital Jin-Shan Branch, No. 7, Yulu Rd., Wuhu Village, Jinshan Dist., New Taipei City 20844, Taiwan; 2Department of Orthopedics, National Taiwan University College of Medicine, National Taiwan University Hospital, No. 1, Section 1, Jen-Ai Road, Taipei 100, Taiwan; hychen83@gmail.com; 3Department of Orthopedic Surgery, Far Eastern Memorial Hospital, No. 21, Section 2, Nanya S. Road, Banciao Dist., New Taipei City 220, Taiwan; orthocch@mail.femh.org.tw; 4Graduate School of Biotechnology and Bioengineering, Yuan Ze University, No. 135, Yuan-Tung Road, Chuang-Li Dist., Taoyuan 320, Taiwan; 5Department of Surgery, National Taiwan University Hospital, No. 7, Chung Shan S. Rd., Taipei 10002, Taiwan; Yywivy@gmail.com; 6Institute of Biomedical Engineering, College of Medicine and College of Engineering, National Taiwan University, No. 1, Section 1, Jen-Ai Road, Taipei 100, Taiwan; image0120@gmail.com; 7Department of Anesthesiology and Pain Medicine, Min-Sheng General Hospital, No. 168, Ching-Kuo Rd., Taoyuan 330, Taiwan; m000924@e-ms.com.tw; 8Department of Materials Science and Engineering, National Taiwan University, No. 1, Section 4, Roosevelt Road, Taipei 10617, Taiwan

**Keywords:** antimicrobials agents, amphiphilic block copolymer, quaternized polymer, hemolysis, micelle

## Abstract

In this study, a novel polystyrene-block-quaternized polyisoprene amphipathic block copolymer (PS-*b*-PIN) is derived from anionic polymerization. Quaternized polymers are prepared through post-quaternization on a functionalized polymer side chain. Moreover, the antibacterial activity of quaternized polymers without red blood cell (RBCs) hemolysis can be controlled by block composition, side chain length, and polymer morphology. The solvent environment is highly related to the polymer morphology, forming micelles or other structures. The polymersome formation would decrease the hemolysis and increase the electron density or quaternized groups density as previous research and our experiment revealed. Herein, the PS-*b*-PIN with *N,N*-dimethyldodecylamine as side chain would form a polymersome structure in the aqueous solution to display the best inhibiting bacterial growth efficiency without hemolytic effect. Therefore, the different single-chain quaternized groups play an important role in the antibacterial action, and act as a controllable factor.

## 1. Introduction

Antibacterial agents are substances that can be added into materials to produce antibacterial properties in order to directly kill bacteria or inhibit the growth of bacteria for a long time. There are significant demands to develop new antibacterial agents while facing severe challenges. Since most antibiotics function by releasing low molecular weight biocide to kill bacteria or inhibit the growth of the bacteria, issues such as (1) rapid revolution of resistance mechanism to cause ineffectiveness, (2) limited capability due to chronic and repeating usage, and (3) environmental problems must be addressed. Consequently, the number of new antibiotics approved for marketing per year declines continuously. Instead of developing new antibiotics, the use of alternative antibacterial agents with different mechanisms to eliminate bacteria, such as silver nanocomposites, host-defense peptides, and synthetic polyelectrolytes, have drawn increasing attention recently [1,2,3,4,5]. Most silver nanocomposites have demonstrated good antibacterial ability; however, high nanosilver concentration may cause harmful environmental impacts, as well as cytotoxicity and genotoxicity in living organisms [6,7]. Host-defense peptides, usually found in the innate immune system, kill bacteria by damaging the cell membrane or invading the cell when approaching bacteria [8,9,10]. These antibacterial peptides generally have low cytotoxicity; however, the expensive production and susceptibility to proteolysis will challenge the success in further commercialization [11,12].

The synthetic polyelectrolytes bearing ionic groups, including phosphor-, sulfo- derivatives, and ammonium groups, have drawn increasing attention recently since they can be manufactured on a large scale with low cost, while killing bacteria in ways similar to peptides [13,14,15]. In addition, the antimicrobial or biological activity can be manipulated by tailoring the molecular structure, such as the molecular weight, density, and type of ionic groups, and even by building random or block copolymers [16,17]. Of the polyelectrolytes, quaternary ammonium groups are the mostly explored of late due to promising bacteria-killing performance and good environmental stability. Nevertheless, the positive antibacterial property is usually accompanied by high cytotoxicity (hemolysis), which becomes the most critical issue for the quaternized polyelectrolytes.

The balance between hydrophobicity and hydrophilicity of the polyelectrolyte is the key to achieving good anti-bacterial activity and low cytotoxicity concurrently, which may be achieved by molecular engineering on the polyelectrolytes. Homopolymers such as polynorbornene can be feasibly synthesized via homopolymerization of designated monomers [18]. However, tuning the hydrophobicity can only be achieved by changing the chemical structure of the polymer backbone, which requires specific molecular engineering on the monomer structure. Tew and colleagues have synthesized four polynorbornene homopolymers with different backbone structures to tailor hydrophobicity, one of which showed distinctive selectivity between anti-bacterial activity and hemolytic property [19]. In contrast to homopolymers, random copolymers can tailor amphiphilicity easily by adopting different monomers in a variety of compositions [20,21,22]. Mizutani et al. have prepared a series of degradable copolymers to examine the effect of polymer properties on their antimicrobial and hemolytic activities. The acrylate copolymer with quaternary ammonium groups and the acrylamide copolymers shows low or no antimicrobial and hemolytic activities [23]. Directly combining the useful monomers with functional groups selectivity is the advantage of random copolymer, but the lack of precise synthesis makes it difficult to control the antibacterial activity, hemolysis, and even cytotoxicity. Sen et al. have synthesized and compared series of amphiphilic pyridinium polymers, and observed that the spatial positioning of the charge and tail significantly influences the toxicity of these polymers. This result may be used as a guiding principle in the design of polymeric antimicrobial compounds with reduced toxicity [24]. Song et al. have tested four series of polymers with cationic and hydrophobic groups distributed along the backbone against six different bacterial species, and for host cytotoxicity. In their study, the antibacterial and hemolytic activities of polymers can be controlled by the exact distance of ammonium groups along the backbone [25].

Compared to homopolymers and random copolymers, block copolymers are more advantageous in controlling the microstructures, which are not well discussed in most homo or random copolymer literature. The antibacterial activity relates to the morphology in solution. By varying the lengths of the constituent blocks or ratio of the amphiphilic blocks, block copolymer could be transformed into a designated molecular structure [26]. Kenichi Kuroda et al. have reported that the block and random copolymers with similar polymer lengths and monomer compositions display the same level of bactericidal activity, but the block copolymers display selective activity against *E. coli* over red blood cells (RBCs) [27]. In a previous study, we synthesized a polystyrene-block-quaternized polyisoprene amphipathic block copolymer, denoted as PS-*b*-PIN for alkaline direct methanol fuel cells [28]. PS-*b*-PIN is amphiphilic block copolymer composed of a hydrophobic polystyrene (PS) segment and a quaternized polyisoprene (PIN) segment bearing pendant quaternary ammonium groups attached to the polyisoprene backbone via alkyl spacers. Since PS-*b*-PIN is amphiphilic block copolymer and contains many of quaternized side chains, it can be used as an antibacterial agent. Herein, we assembled the prepared PS-*b*-PIN with different long side chains into large polymersomes in the buffer solution to serve as antibacterial agents. Through precise synthesis of block copolymer, the antimicrobial activities can be controlled and polymer morphology manipulated to examine the relationship between them.

## 2. Materials and Methods

### 2.1. Materials

PS-*b*-PIN with different long side chains was provided by author Chi-Yang Chao from the Department of Materials Science and Engineering at National Taiwan University (NTU). Tetrahydrofuran (THF, Reagent Grade) was dried from a mixture of sodium under nitrogen atmosphere. Ethanol (≥99.5%), dimethyl sulfoxide (DMSO), human hemoglobin, Triton X-100, and 3-[4,5-dimethylthiazol-2-yl]-2,5-diphenyl tetrazolium bromide (MTT) were purchased from Sigma-Aldrich (St. Louis, MO, USA).

### 2.2. Surface Charge Determination of PS-b-PIN

The PS-*b*-PIN stock solutions with different side chain length were diluted by ddH_2_O to 1000 ppm. The zeta potential of the prepared PS-*b*-PIN solutions were determined using a Zetasizer Nano-ZS90 (MalvernInstruments Ltd., Malvern, UK), based on laser Doppler electrophoresis.

### 2.3. Antibacterial Analysis of PS-b-PIN

The antimicrobial efficacy of PS-*b*-PIN was investigated by using pathogenic bacterial strains of *Escherichia coli* (American Type Culture Collection (ATCC) 25922) and S*taphylococcus aureus* (ATCC 23235), obtained from the Department of Clinical Laboratory, Sciences and Medical Biotechnology at National Taiwan University (Taipei, Taiwan). Measurements of bacterial growth were obtained following the protocol of the ISO 22196 standard test methods. Typically, bacteria were cultivated in 3% Bacto tryptic-soy broth (TSB) (Becton Dickinson, Sparks, MD, USA) at 37 °C for 12 h. After serial dilution of the suspension, an aliquot of solution (2 mL) was spread on Luria-Bertani (LB) agars and incubated at 37 °C for 12 h. The bacterial concertation of the suspension was decided by colony counting assay with approximately 1 × 10^5^ colony formation unit per milliliter (CFU/mL). The bacteria suspension was then diluted into 1 × 10^5^ CFU/mL by TSB for later antibacterial tests.

To test the effect of side chain length of PS-*b*-PIN on antibacterial, the PS-*b*-PIN stock solutions with different side chain length were prepared at the concentration of 5000 ppm in 25% (*w*/*w*) ethanol aqueous solution and well dispersed by a sonication process for over 30 min before use. The TSB diluted PS-*b*-PIN solutions were then mixed with the *E. coli* with equal volume of 250 μL in separate micro tubes with final PS-*b*-PIN concentration of 0.4, 2, 10, 50 and 100 ppm. After orbital shaking incubation (37 °C, 30 rpm) for 16 h, the OD of TSB solutions containing *E. coli* were determined by ultraviolet-visible (UV-Vis) spectrophotometer (Cary 50 Conc; Varian, Palo Alto, CA, USA) at 600 nm. Sterile and inoculated culture media were used as a negative and positive control, respectively. The growth rate of *E. coli* was calculated according to the following equation:% *E. coli* growth rate = 100 − (OD sample/OD positive control) × 100%

Moreover, the TSB diluted PS-*b*-PIN solutions were mixed with *E. coli* or *S. aureus* solution with equal volume of 250 μL in separate micro tubes with a final PS-*b*-PIN concentration of 100, 1000, and 2500 ppm. After orbital shaking incubation (37 °C, 30 rpm) for 3 and 5 h, bacterial suspensions were collected from the tube and then stained by BacLight LIVE/DEAD bacterial viability kit (Thermo Fisher Scientific, Waltham, MA, USA) following the manufacturer’s instructions. The resulting samples were observed under a fluorescence microscope. Live and dead bacterial cells were counted through image analysis by ImageJ software (National Institutes of Health, Bethesda, MD, USA).

### 2.4. Cytotoxicity Test of PS-b-PIN

NIH 3T3 cells, derived from NIH Swiss mouse embryo cultures, were cultured in Dulbecco’s modified Eagle’s medium (DMEM) supplemented with 10% fetal bovine serum (FBS) and penicillin and streptomycin (100 μg/mL) at 37 °C in an atmosphere of 5% CO_2_, and the culture medium was changed on alternate days.

NIH 3T3 cells were seeded into 48-well plates at the density of 1 × 10^3^ cells/well and incubated for 24 h. The culture medium was pre-mixed with the prepared PS-*b*-PIN stock solution with final concentration of 1, 10, 100, 1000, and 2500 ppm for 24 h at 37 °C, and then followed by centrifugation at 2800 rpm for 5 min. The supernatant was then applied for culturing of pre-seeded NIH 3T3 cells for 24 h.

For cell viability assay, 0.5 mg/mL MTT solution was added to each well and incubated for 3 h at 37 °C. Subsequently, the MTT solution was aspirated, and the formed formazan crystals were dissolved in DMSO. The spectrophotometric absorbance at 570 nm was measured using a multi-well plate reader (PowerWave X, BioTek Instruments, Winooski, VT, USA).

### 2.5. Hemolysis Test of PS-b-PIN

Hemolytic activity was evaluated by examining hemoglobin release from rat red blood cells (RBCs) by direct contact to PS-*b*-PIN. Fresh rat whole blood was collected in spray-dried K2EDTA tube (Thermo Fischer Scientific, Waltham, MA, USA) and tested within three days. Next, 180 μL of hole blood was mixed with PS-*b*-PIN and 100 μL phosphate buffered saline buffer (PBS) with final concentration of 100, 1000, and 2500 ppm by totally volume of 300 μL in micro centrifuge tubes for 2 min. The RBCs in the mixed solution were then washed by repeatedly add of 1000 μL PBS, centrifugation (1200 rpm, 4 °C for 5 min) and supernatant was removed; this was repeated five times. The remaining RBCs were lysed to release of hemoglobin by adding of 800 μL H_2_O and centrifuged under 3000 rpm at 4 °C for 5 min. Then, 10 μL of solution was mixed with 90 μL of Drabkin’s reagent (Sigma-Aldrich, St. Louis, MO, USA) to oxidize the hemoglobin into cyanmethemoglobin and absorbance at 540 nm was measured using a microplate reader (PowerWave X, BioTek Instruments, Winooski, VT, USA). For standard calibration of the absorbance, human hemoglobin and Triton X-100 were used as negative and positive controls of the hemolysis test. Samples with a hemolysis ratio less than 5% were regarded as “no hemolysis” [27].

### 2.6. Morphology Determination by Transmission Electron Microscopy

PS-*b*-PIN were re-dissolved in ethanol/THF (*v*/*v* = 95/5) solution under ultrasonic oscillator for 24 h. A drop of the polymer solution was dipped on a copper mesh and vacuum dried for 24 h. The sample on the mesh was treated with ruthenium tetroxide (RuO_4_) vapour for 3 min to stain the polystyrene rich domains. Transmission electron microscopy (TEM) images were obtained from Hitachi H-7100 Transmission Electron Microscope (Hitachi High-Technologies, Tokyo, Japan) equipped with a CCD camera using an accelerating voltage of 75 keV.

### 2.7. Statistical Analysis

All the data were presented as the mean ± standard deviation. Student’s *t*-test was used to analyze the significance of the differences between groups. The value of *p* < 0.05 was considered statistically significant.

## 3. Results

### 3.1. Zeta Potential of PS-b-PIN

The previously developed PS-*b*-PIN diblock copolymer was synthesized according to the route depicted in Figure 1, provided by author Chi-Yang Chao from the Department of Materials Science and Engineering at National Taiwan University (NTU) [28]. The molecular weight of PS and PI in PS-*b*-PIBr was 6000 and 2000 g/mol, respectively, with the polydispersity index (PDI) of 1.20. Herein, the side chain length in PS-*b*-PIN could be adjusted by reacting PS-*b*-PIBr with trimethylamine (PS-*b*-PIN (C1-Br)), *N,N*-dimethylbutylamine (PS-*b*-PIN (C4-Br)), *N,N*-dimethyloctylamine (PS-*b*-PIN (C8-Br)), *N,N*-dimethyldodecylamine (PS-*b*-PIN (C12-Br)), *N,N*-dimethylhexadecylamine (PS-*b*-PIN (C16-Br)), or *N,N*-dimethylhexadecylamine (PS-*b*-PIN (C18-Br)). Figure 2 shows the zeta potential of PS-*b*-PIN polymer with different side chain lengths in an aqueous solution. Increasing the side chain length increased the zeta potential of PS-*b*-PIN particles, as the carbon side chain length was lower than 12. When the PS-*b*-PIN was conjugated with *N,N*-dimethylhexadecylamine (C16-Br) or *N,N*-dimethylhexadecylamine (C18-Br), there was no significant difference in zeta potential in comparison with that conjugated *N,N*-dimethyldodecylamine (C12-Br).

### 3.2. Effect of Side Chain Length of PS-b-PIN on Antibacterial

In order to test the effect of side chain length of PS-*b*-PIN on inhibition of bacterial growth, the prepared PS-*b*-PIN (C1-Br), PS-*b*-PIN (C4-Br), PS-*b*-PIN (C8-Br), PS-*b*-PIN (C12-Br), PS-*b*-PIN (C16-Br), and PS-*b*-PIN (C18-Br) at the concentrations of 0.4, 2, 10, 50, and 100 ppm were incubated with *E. coli* for 16 h. The results of *E. coli* growth rate shown in Figure 3 reveal that there was a positive relationship between the bacterial growth rate and PS-*b*-PIN concentration. Moreover, the antibacterial effect of PS-*b*-PIN increased as the length side chain increased. However, the PS-*b*-PIN with *N,N*-dimethyldodecylamine as a side chain could achieve a better antibacterial effect due to the higher surface charge density, which could lead to ionic interaction with bacteria wall constituents and disruption of the cell bacteria membrane to cause bacterial death [29,30]. However, the PS-*b*-PIN with longer side chain length (C16-Br and C18-Br) could not exhibit an effective inhibition on the *E. coli* growth. It could be suggested that more flexible long carbon side chain would form intramolecular aggregates to reduce the active polymer chains, resulting in the lower antibacterial activity. Since the PS-*b*-PIN (C12-Br) had best effect on inhibiting bacterial growth, we decided to use this kind PS-*b*-PIN block copolymer sample for the following tests.

To investigate the antibactericidal ability of PS-*b*-PIN against both Gram-negative and Gram-positive bacteria, 1 × 10^5^ CFU/mL of bacteria were directly mixed with PS-*b*-PIN under orbital shaker incubator at 37 °C. In order to examine the antibacterial mechanism of PS-*b*-PIN, the bacteria number with different treatment were assayed at the incubation time of 3 and 5 h (Figure 4). Since PS-*b*-PIN has an absorption background at 600 nm under tested concentration, Live/Dead fluorescence assays were used. After 3 h of incubation, the number of *E. coli* was lower for control and PS-*b*-PIN treated groups. This was due to the lag phase of bacterial growth in which the number increase of bacteria was merely assayed. After 5 h incubation, *E. coli* started to grow in log phase and an obvious number increase was observed for the control group. In contrast, growth of *E. coli* was clearly inhibited by mixing with PS-*b*-PIN within a 5 h incubation. Similar growth tendency was also observed for *S. aureus*. At the concentration of 2500 ppm, the prepared PS-*b*-PIN against *E. coli* was more efficient than that against *S. aureus.* This greater antimicrobial efficiency might be attributed to the thickness of bacterial wall. The cell wall thickness of Gram-negative bacteria *E. coli* is around 7–8 nm, which was thinner and consequently more susceptible than that of Gram-positive bacteria *S. aureus* (20–80 nm) [31]. These results demonstrated that PS-*b*-PIN has antibacterial activity against not only Gram-negative but also Gram-positive bacteria in the short treating time.

### 3.3. Cytotoxicity of PS-b-PIN

As the pilot study of biocompatibility tests, cytotoxicity was investigated through MTT assays. Since PS-*b*-PIN can be well dispersed in aqueous solution, extract method was used [32]. Therefore, extract solution of PS-*b*-PIN was then used for the culturing of NIH 3T3 cells. Figure 5 shows the cell viability of NIH 3T3 cells incubated with PS-*b*-PIN extract solution for 24 h. It was observed that for PS-*b*-PIN with extracted concentration from 100 to 1000 ppm, over 80% of cell viability was reached. Since no cell toxicity dependency on PS-*b*-PIN concentration was observed, it was assumed that the copolymer chain of PS-*b*-PIN was stable in the aqueous phase and no toxic ingredients were dissolved in the solution.

### 3.4. Hemocompatibility of PS-b-PIN

The amphiphilic copolymer, PS-*b*-PIN, was further investigated with the breakage ability toward lipid bilayer of cell membrane. For that, lytic activity of the PS-*b*-PIN copolymer against RBCs was examined through hemolysis assays. By directly mixing copolymer with rat whole blood, PS-*b*-PIN showed no hemolysis (<5%) at all tested concentrations (100–2500 ppm). It is also noteworthy that the hemolysis does not show concentration dependence on copolymer (Figure 6). In contrast to PS-*b*-PIN, the non-ionic surfactant, Triton X-100, was observed to induce 95% hemolysis of RBCs. It is known that non-ionic surfactants such as Triton can exist as both micelles and monomers in solution and interfere with protein-lipid and lipid-lipid interaction of cell membrane as lysis agents [33]. On the other hand, the prepared PS-*b*-PIN, as an amphiphilic functional group, is theoretically stronger lysis agent for RBCs membranes, as it did for NIH 3T3 cell membranes [34]. The ability to destroy the RBCs and NIH 3T3 cell membranes might be attributed to the difference in the membrane structure and composition. It is well known that the RBCs has a double-layer phospholipid structure with a thickness of approximately 10 nm, in which many protein molecules intercalate on the membrane, and there is a strong net-like fiber tissue under the membrane to support the membrane structure [35]. Therefore, these intercalated proteins and net-like fiber tissue would provide RBCs with greater flexibility and deformability than NIH 3T3 cells against the PS-*b*-PIN destruction.

### 3.5. Morphologies of PS-b-PIN Polymersome

To examine the morphology of PS-*b*-PIN, 5000 ppm of PS-*b*-PIN was dissolved in ethanol/THF (*v*/*v* = 95/5) solution within an ultrasonic oscillator bath for one day. One drop of the polymer solution was dipped onto a copper mesh for TEM visualization observation. The PS-*b*-PIN was dried and treated with RuO_4_ vapour to selectively stain the PS domains, which appeared dark in the TEM image, as shown in Figure 7. The hollow sphere with a diameter of 0.5–1 μm was clearly observed, and the thickness of a circle was around 80 nm for each sphere regardless the diameter. The image accordingly suggests that PS-*b*-PIN could form microstructure in solution. Since PS segment is hydrophobic and PIN segment is hydrophilic, PS-*b*-PIN would form large polymersomes in the buffer solution with PIN segments facing out by its hydrophilic interaction.

When a polymersome forms in the aqueous solution, the hydrophobic chains (PS segments) interfold inside, and positively charged quaternary ammonium chains (PIN segments) are exposed on the surface. On the other hand, the bacterial cell wall surface is negatively charged due to teichoic acids and lipopolysaccharides of Gram-positive and Gram-negative bacteria, respectively [36,37]. The polymersome then tends to interact and attach to the bacterial surface through charge-charge interaction. Since quaternary ammonium chains contain long alkyl structures, they can interrupt the bacterial cell membrane by inserting the alkyl chains into the lipid-lipid bilayer. As the size of the PS-*b*-PIN polymersome is approximately 500 nm, it can tear down the bacteria (about 3–4 μm) while maintaining its polymersome structure. After reacting with bacteria, the polymersome can disperse back into the solution again for another bactericidal function (Figure 8). Therefore, PS-*b*-PIN polymersome can kill bacteria while dispersed into the solution and also maintains a low bacterial number during a certain incubation time. However, PS-*b*-PIN may aggregate due to electrolytes and proteins in the physiological environment after a lengthy incubation time with bacteria.

## 4. Conclusions

In this study, we have investigated the effects of PS-block-PIN quaternized amphiphilic block copolymer on antibacterial activity. The amphiphilic block copolymer structure is a significant element in our research. Through molecular composition and morphology control, we were able to manipulate the morphology into polymersome in the solution. Due to the polymersome structure, it is possible to strike a balance among antibacterial, hemolytic, and cytotoxic activities. The block copolymer displayed high bactericidal activity against *E. coli*, while it displayed selective activity over RBCs and was not hemolytic. So far, we have manipulated the polymer structure to attain good antibacterial activity. This study highlights quaternized amphiphilic block copolymer structures as a new design agent to improve the activity by quaternization on side chains, as well as to understand the mechanism of antibacterial actions by the formation of morphology.

## Figures and Tables

**Figure 1 polymers-14-00250-f001:**
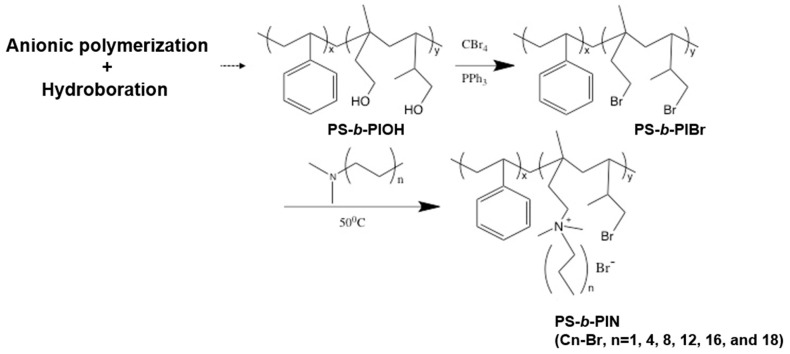
Synthesis of PS-*b*-PIN block copolymers with different side chain length.

**Figure 2 polymers-14-00250-f002:**
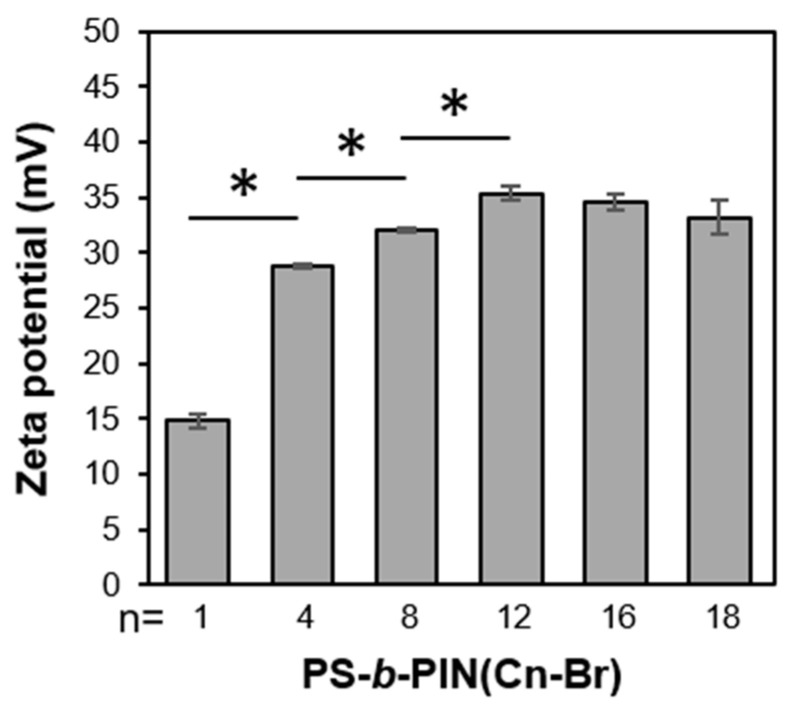
The zeta potential of PS-*b*-PIN with different side chain length. *: *p* < 0.05.

**Figure 3 polymers-14-00250-f003:**
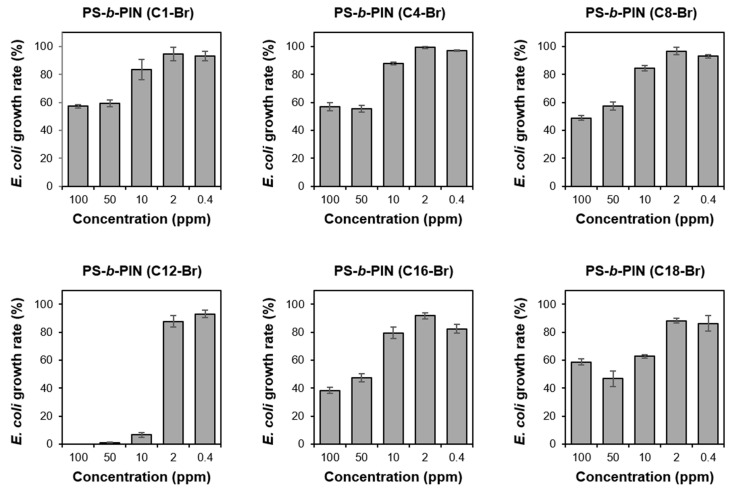
The *E. coli* growth rate after treatment with PS-*b*-PIN with different side chain lengths for 16 h.

**Figure 4 polymers-14-00250-f004:**
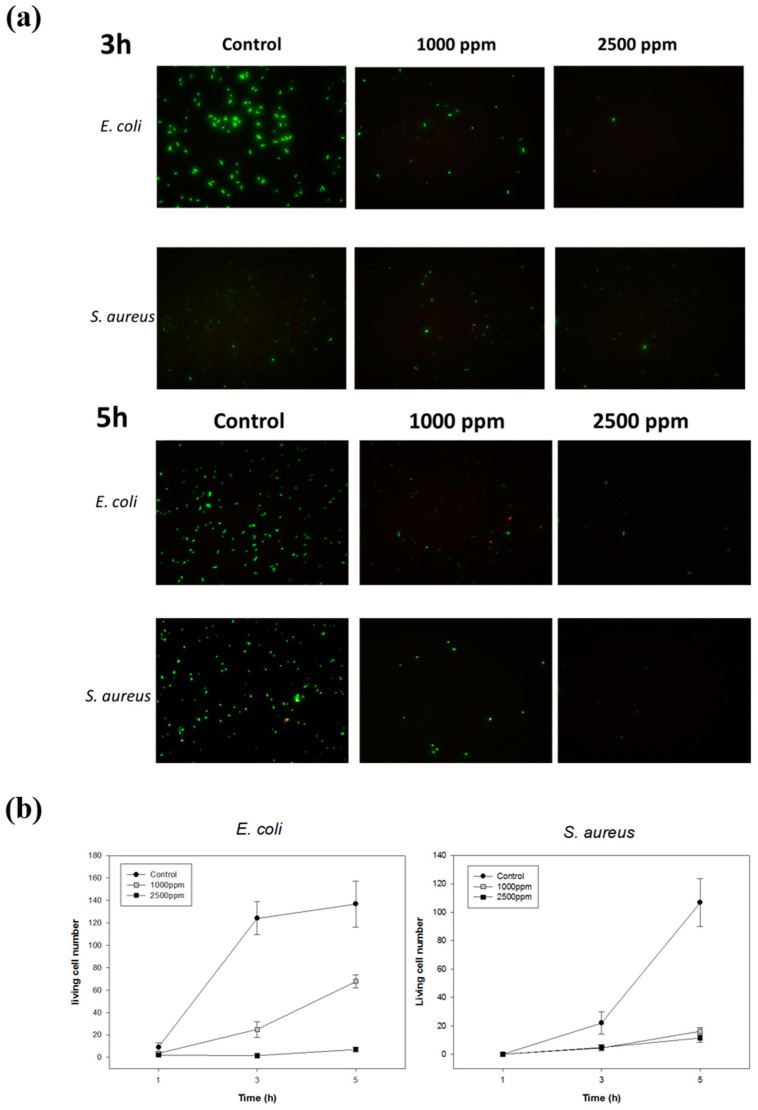
Inhibition of bacterial growth at different PS-*b*-PIN concentrations. (**a**) Fluorescence images and (**b**) the living number of *E. coli* and *S. aureus* with different PS-*b*-PIN concentrations treatment at 3 and 5 h.

**Figure 5 polymers-14-00250-f005:**
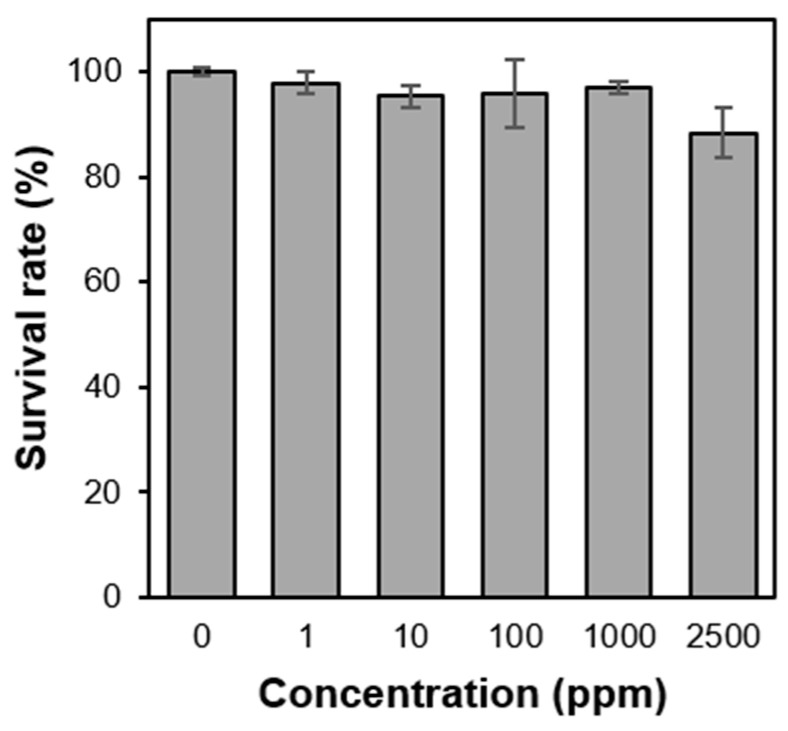
The cytotoxic effect toward 3T3 cells at different PS-*b*-PIN concentration.

**Figure 6 polymers-14-00250-f006:**
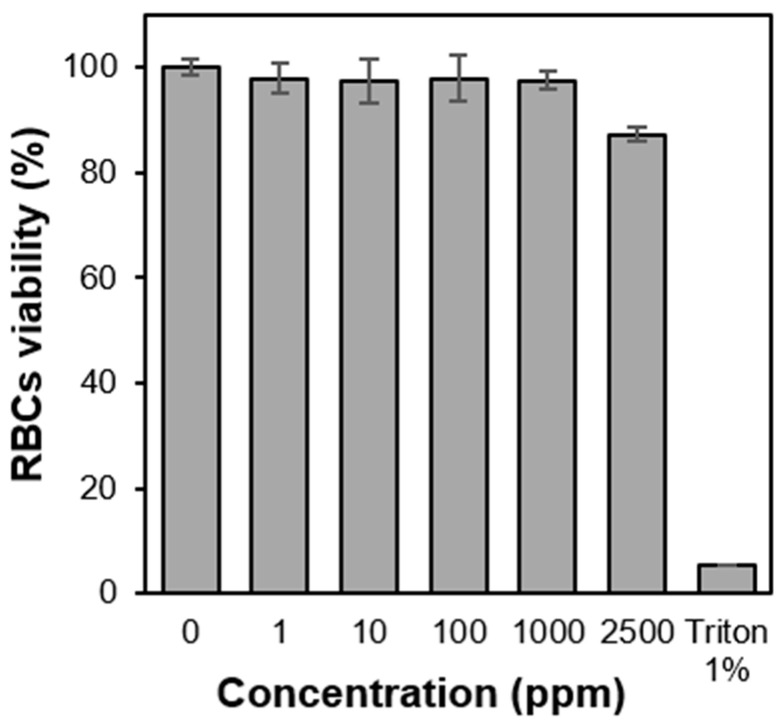
Hemolysis test of PS-*b*-PIN on RBC cells.

**Figure 7 polymers-14-00250-f007:**
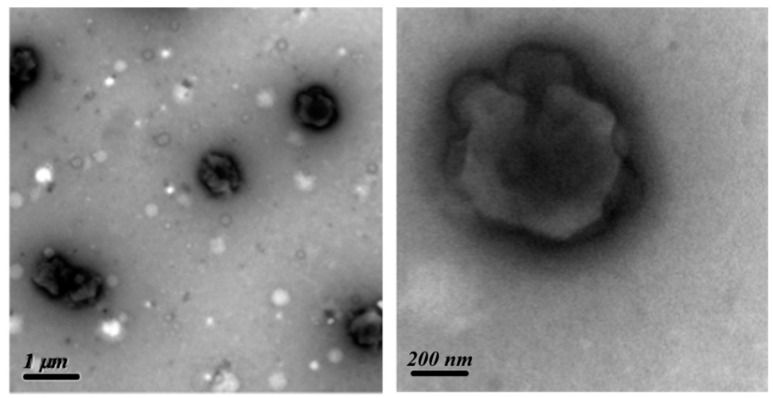
TEM images of PS-*b*-PIN with RuO_4_ stained.

**Figure 8 polymers-14-00250-f008:**
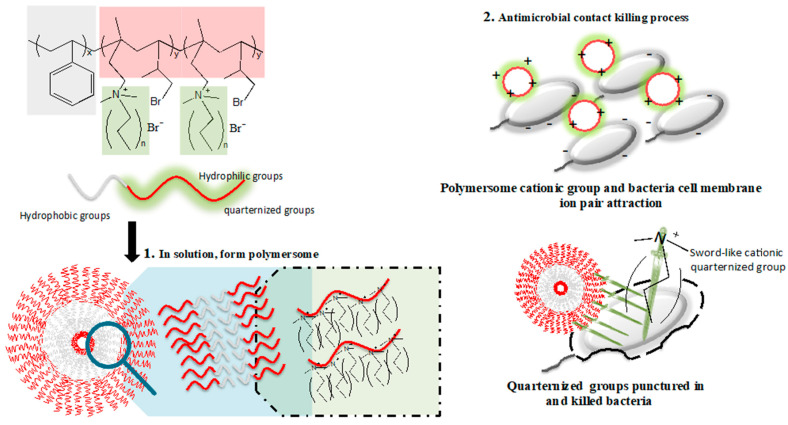
Schematic of proposed quaternized amphiphilic block copolymers PS-*b*-PIN polymersome formation and antibacterial activities.

## Data Availability

The data presented in this study are available on request from the corresponding author.

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
