# Peer review of "Quaternized Amphiphilic Block Copolymers as Antimicrobial Agents"

_polymers, 2022, doi:10.3390/polym14020250_

Round 1

Reviewer 1 Report

Authors present an interesting paper on the employment of amphiphilic block copolymer (PS-PIN) for antibacterial purpose.

There is a mistake in the tittle: ....as antimicrobialS agents should be corrected

At the abstract, they write "block copolymer of poly(styrene)" and could be confussing. It is better to write the whole name of the copolymer for its identification through the text.

Moreover, the sentence at lines 25-26 is very confussing and should be rewritten. The same with the sentence at line 31.

The introduction is well presented and several aspects related to their research are exposed, showing the research with antibacterial purposes developed by several authors with silver nanoparticles, synthesized polyelectrolytes bearing ionic groups, homopolymers or random copolymers and finally, block copolymers. However, their PS-b-PIN copolymer is first mentioned at that point, and should be defined before (at the abstract, for example, as suggested). Perhaps a slightly longer discussion on block copolymers and structures that could obtain should be included.

Regarding materials and methods, they claim that the block copolymer was supplied by one of the authors without any mention on the synthesis procedure or characteristics such as chaing lengths, etc. This point should be improved.

Results and discussion are well presented and interesting test performed. However, the presentation of results is quite confussing and the order in which the captions, figures and their citing text appear should be improved. In lines 206 and 207, there are the captions of figures, should be properly placed. The graph on Z potential is cited and commented in the text well before showing it and could be confussing. The same for the caption of figure 3 in line 224.

However, the test performed seem to be  correct, together with the scheme in which they show the formation of polymersomes and their interaction.

The only question could be, as they claimed for a study on the side chain  length, but there is no morphological evidence on that, regarding the microstructures obtained with different copolymers....

Conclusions are supported by obtained data

Reviewer 2 Report

The paper submitted by Chang et al. deals with the characterization of some block copolymers based on PS-PI as antimicrobials agents. Even if their synthesis have already been published, in this manuscript the authors investigated their antimicrobial properties as a function of their molecular characteristics. 

The manuscript is clear, well written and the conclusions are supported by the results. However, some corrections are needed before publication:

  1. the abstract section is too general. the authors must indicate more specific results.
  2. even if the synthesis of these copolymers has published before, the authors must indicate their molecular characteristics (Mn, dispersity).
  3. there are some errors concerning the formatting of the caption of figure 2 and fig 6 is placed at the wrong place
  4. L223: to use this PS-b-PIN copolymer sample
  5. L238: replace "efficiency" with "efficient"
  6. L289: replace "aquaria" with "aqueous"
  7. the first sentence from the conclusion section must be corrected. the authors have not synthesized in this study the block copolymer samples but they only studied their antibacterial activity.
  8. L310: these copolymer samples form polymersomes and not micelles. please take this aspect into consideration in all the manuscript.
